# Sex-Specific Impact of Fkbp5 on Hippocampal Response to Acute Alcohol Injection: Involvement in Alterations of Metabolism-Related Pathways

**DOI:** 10.3390/cells13010089

**Published:** 2023-12-31

**Authors:** Kent E. Williams, Yi Zou, Bin Qiu, Tatsuyoshi Kono, Changyong Guo, Dawn Garcia, Hanying Chen, Tamara Graves, Zhao Lai, Carmella Evans-Molina, Yao-Ying Ma, Suthat Liangpunsakul, Weidong Yong, Tiebing Liang

**Affiliations:** 1Division of Gastroenterology and Hepatology, Department of Medicine, Indiana University, Indianapolis, IN 46202, USA; kew5@iu.edu (K.E.W.); tharris1@iu.edu (T.G.); sliangpu@iu.edu (S.L.); 2Greehey Children’s Cancer Research Institute, The University of Texas Health Science Center at San Antonio, San Antonio, TX 78229, USA; zou@uthscsa.edu (Y.Z.); datgarcia5@sbcglobal.net (D.G.); laiz@uthscsa.edu (Z.L.); 3Department of Pharmacology, Yale University School of Medicine, New Haven, CT 06520, USA; kennyqiu@live.com; 4Diabetes Research Center, Division of Endocrinology, Department of Medicine, Indiana University School of Medicine, Indianapolis, IN 46202, USA; konot@iu.edu (T.K.); cevansmo@iu.edu (C.E.-M.); 5Department Pharmacology and Toxicology, Indiana University School of Medicine, Indianapolis, IN 46202, USA; guo14@iu.edu (C.G.); ym9@iu.edu (Y.-Y.M.); 6Department Medical and Molecular Genetics, Indiana University School of Medicine, Indianapolis, IN 46202, USA; hanchen@iu.edu; 7Roudebush Veterans Administration Medical Center, Indianapolis, IN 46202, USA; 8Department of Biochemistry and Molecular Biology, Indiana University School of Medicine, Indianapolis, IN 46202, USA; 9Department of Surgery, Indiana University School of Medicine, Indianapolis, IN 46202, USA; yongwd@hotmail.com

**Keywords:** RNA-seq, hippocampus, Fkbp5, mitochondria, lipid metabolism

## Abstract

High levels of alcohol intake alter brain gene expression and can produce long-lasting effects. FK506-binding protein 51 (FKBP51) encoded by *Fkbp5* is a physical and cellular stress response gene and has been associated with alcohol consumption and withdrawal severity. *Fkbp5* has been previously linked to neurite outgrowth and hippocampal morphology, sex differences in stress response, and epigenetic modification. Presently, primary cultured *Fkbp5* KO and WT mouse neurons were examined for neurite outgrowth and mitochondrial signal with and without alcohol. We found neurite specification differences between KO and WT; particularly, mesh-like morphology was observed after alcohol treatment and confirmed higher MitoTracker signal in cultured neurons of *Fkbp5* KO compared to WT at both naive and alcohol-treated conditions. Brain regions that express FKBP51 protein were identified, and hippocampus was confirmed to possess a high level of expression. RNA-seq profiling was performed using the hippocampus of naïve or alcohol-injected (2 mg EtOH/Kg) male and female *Fkbp5* KO and WT mice. Differentially expressed genes (DEGs) were identified between *Fkbp5* KO and WT at baseline and following alcohol treatment, with female comparisons possessing a higher number of DEGs than male comparisons. Pathway analysis suggested that genes affecting calcium signaling, lipid metabolism, and axon guidance were differentially expressed at naïve condition between KO and WT. Alcohol treatment significantly affected pathways and enzymes involved in biosynthesis (Keto, serine, and glycine) and signaling (dopamine and insulin receptor), and neuroprotective role. Functions related to cell morphology, cell-to-cell signaling, lipid metabolism, injury response, and post-translational modification were significantly altered due to alcohol. In summary, *Fkbp5* plays a critical role in the response to acute alcohol treatment by altering metabolism and signaling-related genes.

## 1. Introduction

Heavy alcohol intake within a short period of time can produce long-lasting damage to the brain [1,2]. Previous research demonstrated that an acute ethanol challenge mimics this high level of alcohol consumption and results in gene expression alterations in the brain, immune system adaptations, and an increased risk of alcohol use disorder (AUD) development [3,4]. Acute alcohol intake stimulates hypothalamic–pituitary–adrenal (HPA) axis regulation, and acute alcohol injection serves as a useful tool to understand the immediate response of brain regions. Glucocorticoid and glucocorticoid receptor (GR) are critical molecules in the regulation of HPA axis and alcohol consumption [5,6,7,8]. The mature GR complex has several major components: GR, heat shock protein 90 (HSP90), and its co-chaperones (e.g., FK506 binding proteins).

FKBP51 (FK506-binding protein 51, encoded by *Fkbp5*) belongs to a subclass of immunophilin proteins and possesses peptidyl-prolyl *cis–trans* isomerase (PPIase) activity that is crucial for protein complex formation, signal transduction, and protein trafficking [9,10,11,12,13,14]. FKBP51 plays a defining role in HPA axis negative feedback regulation by reducing GR activity [15,16], and directly modulates the stress response to various insults, both physical and chemical [17,18,19]. Both GR and FKBP51 are associated with alcohol use disorder [20,21] and *Fkbp5* has also been associated with drug addiction and alcohol use [21,22,23,24,25], with its expression altered following alcohol and drug administration [26,27,28]. We previously demonstrated that *Fkbp5* KO mice consume more alcohol and possess enhanced sensitivity to alcohol withdrawal [21,22]. The selective inhibition of FKBP51 has been discussed as a potential approach in the treatment of anxiety, alcohol use, cocaine addiction, and depression [29,30,31].

The hippocampus is a central contributor to both cognitive function and alcohol consumption behavior, although alcohol consumption can compromise hippocampal function through glucocorticoid and GR regulation [5,20,32,33]. Hippocampal output also contributes to the inhibition of the HPA-axis [34]. Binge, chronic, or prenatal alcohol consumption is associated with significantly lower hippocampal volume in humans [35,36,37,38,39]. Previous research from our lab and others revealed that *Fkbp5* is highly expressed in the basolateral amygdala (BLA) and in the hippocampal sub regions CA2, CA3, and dentate gyrus (DG) [21,40]. Recently, we demonstrated that the lack of FKBP51 modulates hippocampal size [41].

Little has been established about the mechanism of *Fkbp5* involvement in acute alcohol insult and its impact at the transcriptomic level in the hippocampus. We compared the alcohol treatment effect on primary cultured neuron between KO and WT. We chose to examine hippocampal gene expression using RNA-seq profiling and identified the pathways altered following alcohol treatment in *Fkbp5* KO and wild-type (WT) mice. Both male and female *Fkbp5* KO and WT mice were used to better understand the sex differences in basal and alcohol-induced hippocampal gene expression.

## 2. Materials and Methods

### 2.1. Animals

All experimental procedures were approved by the Indiana University School of Medicine Institutional Animal Care and Use Committee and by the Chinese Academy of Medical Sciences, the Institute of Laboratory Animal Sciences committee in accordance with the guidelines of the Guide for the Care and Use of Laboratory Animals. The animals were maintained in facilities fully accredited by the Association for the Assessment and Accreditation of Laboratory Animal Care (AAALAC). Generation of *Fkbp5* knockout (*Fkbp*5 KO) mice was described in a previous publication (Yong et al. 2007 [54]). *Fkbp*5 KO and WT littermates were bred through heterozygous mating on a C57BL/6J background and backcrossed at least 10 generations. All animals were maintained on regular chow diet, with a 12-h light/dark cycle from 7:00 am–7:00 pm.

### 2.2. Primary Postnatal Neuron Culture

Primary cultures of hippocampal neurons were prepared from mouse embryos at embryonic E15.5 day (E15.5) or postnatal day 0 (P0), as previously described with modification [41,42]. The hippocampus was dissected following published procedures, and dissected tissues were subjected to digestion with 0.25% Trypsin (Thermo Fisher Scientific, Inc., Waltham, MA, USA) for 15 min at 37 °C, prior to the addition of FBS to halt digestion. Cells cultured for immunofluorescence were plated to a concentration of 2 × 10^4^ cells per well of 24-well plates containing Poly-L-Lysine-treated glass coverslips. Cells were treated every 2–4 days with Neurobasal medium containing B-27 supplement (Thermo Fisher Scientific). In alcohol-treated cultures, a final concentration of 100 mM EtOH in the medium was used and maintained in a chamber containing a reservoir of 200 mM EtOH. The experiment was repeated, and each time 2–3 mice of each genotype were used. However, the sex of the embryonic mouse could not be determined visually.

### 2.3. Immunocytochemistry and Quantification

To detect mitochondria, the specific cell-permeable dye 25 nM MitoTracker Red CMXRos (Thermo Fisher Scientific) was incubated with the cells for 30 min at 37 °C before fixing with ice-cold methanol containing 3% acetic acid for 10 min. Cells were washed three times with ice-cold PBS and blocked for 1 h with 10% normal goat serum in PBS plus 0.1% Tween-20. Immunofluorescence of primary cultured neurons on day 3 in vitro (DIV 3) was performed using chicken anti-MAP2 (1:5000, ab5392, Abcam, Plc, Cambridge, UK), rabbit anti-βIII-tubulin (1:500, 5568, Cell Signaling Technology, Danvers, MA, USA), appropriate Alexa Fluor-conjugated secondary antibodies (1:500, A-11040 and A-11034, Thermo Fisher Scientific), and Fluoromount-G with DAPI (Thermo Fisher Scientific), as previously described [43]. All experiments were individually repeated 3–5 times.

Photomicrographs of fluorescently-labeled neurons were collected using an Olympus2 confocal microscope and Fluoview Imaging software (ver. 4.2b) (Evident Corporation, Tokyo, Japan). Primary, secondary, and tertiary neurites were counted based on observed branching patterns. To assess mitochondrial abundance, composite images were split into green and red images, reflecting the signal from the AlexaFluor-488-labeled β3-tubulin and the Mitotracker Red CMXRos, respectively. Green images were used to determine the area of the neuron and red images were used to determine the area of mitochondrial signal. ImageJ was used to adjust the image color threshold brightness and images were converted to binary. All signals outside the area of interest were cropped and despeckling was employed to reduce noise. Particles were measured and their areas were summed to produce the respective neuronal and mitochondrial areas. Mitochondrial density values represent the proportion of the mitochondrial signal to the neuronal signal.

### 2.4. β-Galactosidase Staining

Age-matched adult male *Fkbp5* KO (N = 16) and female (N = 17) mice were used for morphological and histological studies. Whole brains were sectioned by hand using a dissection matrix and fixed in 10% neutral-buffered formalin for 30 min followed by incubating at 37 °C overnight in a pre-warmed solution of 1 mM X-gal, 3 mM potassium ferricyanide, 3 mM potassium ferrocyanide, 0.1% Tx-100 in PBS (pH 7.8) [44].

### 2.5. Alcohol Injection

Male and female mice from each genotype, WT (N = 16) and KO (N = 16), were randomly assigned to receive an injection of alcohol (8 male and 8 female) or naive (8 male and 8 female) of both WT and KO genotypes. Pure ethanol was mixed in saline (0.9% sodium chloride) to make a 12.5% ethanol solution (volume/volume, *v*/*v*) for injection [45]. For the alcohol injection group, a dose of 2 g ethanol per kg body weight (BW) (2 g/kg BW) was used, as it elevates blood alcohol content (BAC) to 200–300 mg/dL [21]; a dose high enough to produce intoxication and sedation in mice selectively bred for alcohol consumption [45].

### 2.6. RNA-Sequencing and Data Analysis

Mice at the average age of 3–4 months old were produced from 2 KO breeding trios and 4 WT breeding pairs of mice. For no alcohol control (naive) groups, mice were randomly assigned to KO-male-naïve (N = 4), KO-female-naïve (N = 4), WT-male-naïve (N = 4), and WT-female-naïve (N = 4); for alcohol groups, the same number of mice were used, they are KO-male-alcohol (N = 4), KO-female-alcohol (N = 4), WT-male-alcohol (N = 4), and WT-female-alcohol (N = 4). These mice were used for hippocampal dissection following the published method [46]. A total of 32 RNA samples were isolated using TRIzol*^®^* followed by RNeasy Mini kit purification (Qiagen, Valencia, CA, USA). Sequencing libraries were constructed using the Illumina TruSeq RNA sample preparation protocol. The resulting libraries were sequenced on an Illumina HiSeq™ 2000 using a standard single-end 50 bp sequencing protocol. The reads were aligned to the reference *Mus musculus* genome (UCSC build MM9) with TopHat [47]. No more than 2 mismatches were allowed in the alignment. HTseq was used to count gene expression (reads) [48], and DEseq [49] was used to find differentially expressed genes (DEGs) after performing median normalization. DEGs were identified with adjusted *p*-values of <0.05 for multiple tests using the Benjamini–Hochberg adjustment [50] and sequence read abundances per gene above 40% within each sample. The significant genes with Fold Change (FC) > 2 were further analyzed. Heatmaps of gene expression levels were generated with R/heatmap2. Genes differentially expressed between KO and WT were analyzed with the functional annotation tool of the DAVID bioinformatics resources 6.7 (http://david.abcc.ncifcrf.gov, accessed on 29 October 2023). Analysis software (IPA, Qiagen, Redwood City, CA, USA) was also applied for the identification of potentially important biological mechanisms and pathways. RNA-seq data have been submitted to a public database (GEO access ID: GSE143354).

### 2.7. Western Blot Analysis

Proteins from hippocampi were harvested in lysis buffer (Beyotime, Nantong, China) with 1:10 volume protease inhibitor (S8800, Sigma-Aldrich, Saint Louis, MO, USA) and 1:100 volume phosphatase inhibitor cocktail (P0044, Sigma-Aldrich) as previously described [51]. Western blotting was performed as previously described [42,51]. Primary antibodies used in this study include rabbit anti-FOLH1 (Sigma-Aldrich, St Louis, MO 1:1000), rabbit anti-PPWD1 (Sigma-Aldrich, St. Louis, MO, USA, 1:1000), and rabbit anti-GAPDH (Santa Cruz, Dallas, TX, USA, 1:5000). Signals were captured using the Tanon 5500 Chemiluminescent Imaging System (Tanon, Shanghai, China) and quantified using TanonGis software (Tanon).

### 2.8. Quantitative Real-Time PCR

Hippocampi were harvested and total mRNA was isolated using TRIzol^®^. Reverse transcription (RT) and real-time PCR were conducted according to the manufacturer’s instruction (TaKaRa Biotechnology Co, Ltd., Dalian, China) using the ABI PRISM 7500 System (Life Technologies, Gaithersburg, MD, USA). In each reaction plate, sample and no template control were included. The relative mRNA expression levels were normalized to endogenous internal control gene *Gapdh* or *Rpl7*, and their expression were not changed by alcohol treatment. GraphPad Prism (ver. 9.5.0) was used for data analysis (GraphPad Software Inc., San Diego, CA, USA), and significance was defined as *p* < 0.05. A list of the primers used can be found in Appendix A.

## 3. Results

### 3.1. Fkbp5 KO Has Enhanced MitoTracker Signal in Cultured Neurons

Previous research identified a role for *Fkbp5* in neurite outgrowth. We assessed the neurite branching (Figure 1) and mitochondrial signal in the presence or absence of alcohol (Figure 2). At day three in vitro (DIV 3), WT hippocampal neurons appeared to possess more primary neurites (5.625 vs. 4.75). However, KO hippocampal neurons appeared to possess more secondary neurites (3.875 vs. 2). Intriguingly, cultured neurons displayed mesh-like alterations in their exteriors following alcohol treatment, which seemed to increase the total number of neurites relative to the non-alcohol groups of both genotypes. We further manually counted cultured neurons and compared the changes in neurite numbers after EtOH (Figure 1B). The average total number of neurites for KO trends higher than WT in the untreated condition, but no differences between the two groups were apparent after EtOH treatment. The changes in neurite number following alcohol exposure revealed that KO possessed fewer tertiary neurites than WT as a percentage of naïve controls at 3 DIV (Figure 1B). FKBP51 is a mitochondrial protein that responds to cellular stress [52,53]. We recently discovered a role for *Fkbp5* in liver mitochondrial function, however, no studies were conducted on the brain at that time. MitoTracker was used to directly assess mitochondrial signal intensity in cultured hippocampal neurons in vitro (Figure 2). KO mitochondrial density appears to be over triple that of WT. Similar trends were also observed in EtOH-treated groups, with KO maintaining a higher density than WT (Figure 2B). In summary, *Fkbp5* KO hippocampal neurons exhibited enhanced neurite outgrowth, but number of tertiary neurite outgrowth were less affected by alcohol in KO than WT neurons. Higher mitochondrial signal was observed in KO, which may signify enhanced mitochondrial function as a potential means of protection from alcohol injury.

### 3.2. Fkbp5 Expression in Brain Regions of Male and Female Mice

Given the presence of the *LacZ* reporter gene in *Fkbp5* KO mice, X-gal staining was used to detect blue coloration in the brain as a surrogate for *Fkbp5* expression [54]. We examined its expression in both sexes and with alcohol treatment, while obtaining better resolution than has been previously reported [40]. Brain areas where *Fkbp5* is highly expressed were clearly identified in both male and female mice without or with alcohol (2 g/kg BW) injection (Figure 3A,B). Among hippocampal subregions, the granular cell layer of DG and pyramidal cells of CA3 possessed the highest expression levels. Ventral medial hypothalamus, amygdala nucleus, and reticular nucleus of thalamus (Rt) are also higher-expressing areas, with enhanced signal in specific subareas and nuclei. Notably, high expression was observed in limbic structures including amygdala, hypothalamus, and hippocampus; areas involved in alcohol addiction. In addition to brain region expression, layer-specific expression patterns may indicate functional connections between regions. A medium level of FKBP51 expression was found in a continuous layer of cortex layer 4, entorhinal cortex (ENT) layer 2/3, piriform area (Pir), piriform amygdala area (PAA), and cortical amygdala area, posterior part, lateral zone (COApl). ENT is a nodal point of cortical–hippocampal cross talk. Lower expression in cortex layer 2/3 was found to connect through ENT layer 2/3 (Figure 3A). RNA quantification confirmed hippocampal upregulation of *Fkbp5* mRNA in WT mice and *Lac*Z reporter mRNA in *Fkbp5* KO mice following EtOH injection (Figure 3C).

### 3.3. Sex Differences in Hippocampal Gene Alterations of KO and WT Mice with and without Alcohol

*Fkbp5* expression in the brain is highly responsive to ethanol treatment [26,27,28] and is associated with alcohol consumption behavior [21,31,55]. To understand the effects of loss of *Fkbp5* on gene expression in the hippocampus, transcriptome profiles of *Fkbp5* KO and WT mice were established by RNA-seq. Both male and female KO and WT mice were randomly assigned to control or alcohol injection treatment groups. Multi-dimensional scaling (MDS) plots provide a visual representation of the pattern of similarity among all studied samples. Alcohol-treated and naïve (No-alcohol) animals were grouped into two distinct sections, revealing the dissimilarity of the naïve and alcohol treatment conditions (Figure 4A). More DEGs between KO and WT were identified in the female comparisons relative to the male comparisons in both alcohol-injected and naïve conditions with *p* < 0.05 (Figure 4B), demonstrating that females exhibit a greater number of DEGs due to *Fkbp5* loss of function. Interestingly, the number of DEGs between KO and WT mice was less pronounced following acute alcohol injection when compared to the naïve condition, suggesting a similar response to alcohol.

### 3.4. Pathway Analysis of Differentially Expressed Genes in KO and WT Mice in Naïve Condition

The number of DEGs (FDR *p* < 0.05) in the hippocampus between KO and WT was plotted (Figure 4A) and pathway analysis was performed to further understand how these DEGs likely affect biological functions and disease. All DEGs (659 genes) in both male and female comparisons were used as input to identify canonical pathways. Top IPA canonical pathways and involved DEGs are shown in Appendix A. Notably, many of the identified pathways are relevant to neuronal function, including calcium signaling, axonal guidance signaling, synaptic Long-Term Depression (LTD), and tyrosine degradation. Identification of the endoplasmic reticulum stress pathway, PPARα/RXRα activation, and the fatty acid β-oxidation I pathway also suggest an important role for *Fkbp5* as a hippocampal regulator of lipid metabolism and associated functions [56,57]. These discoveries highlight the specific molecules altered between KO and WT in these pathways (Appendix A), which warrant further investigation. The top functions affected by loss function of *Fkbp5* include endocrine system development and function, lipid metabolism, small molecule biochemistry, and cell morphology and cell-to-cell signaling (Appendix A). Cell and organ morphology are significantly affected by these DEGs, a finding consistent with previously observed effects of *Fkpb5* on dendritic growth and hippocampal morphology [41,58,59].

We next focused on 118 overlapping genes between the male and female comparisons in the naïve condition to understand changes common to both sexes (Figure 5A). A heat map was utilized to depict the gene expression pattern of the overlapping DEGs between KO and WT in the naïve condition (Figure 5B), which identified four distinct gene clusters (1–4). Genes expressed lower in KO were sorted into clusters 1 and 2, while those expressed higher in KO were sorted into clusters 3 and 4. The most significant functional categories of these genes are indicated on the right side of heatmap, and genes involved in these functions were color-coded to indicate their expression patterns as determined by qRT-PCR. It is worth emphasizing that alterations in some canonical pathways were not previously linked to *Fkbp5* (Figure 5C). Three pathways are above or close to the threshold for significance (*p* < 0.05), with ratios of 0.33, 0.34, and 0.25 for methylglyoxal degradation I, Glycero-3-phosphate shuttle, and fatty acid β-oxidation III, respectively. Methylglyoxal is a toxic compound produced through glycolysis and the metabolism of fatty acids and proteins. Multiple enzymes function to catalyze the conversion of methylglyoxal to a less toxic product. One of them is glyoxalase (encoded by *Glo1*), which is 2-fold higher in KO than WT (Appendix A). Fatty acid β-oxidation occurs in mitochondria and converts fatty acids to acetyl-CoA. Several genes related to Co-A-binding (*Dbi*, *Acadl*, and *Decr2*) were also differentially expressed in KO (Figure 5B). The calcium transport I pathway is enriched in KO and may also affect mitochondrial and neuronal function (Figure 5C) [60].

The functions of these DEGs were predicted by IPA and included both known and novel functions (Figure 5D). Cancer and fibrogenesis function were predicted to be significantly activated. Processing of RNA is noteworthy, as FKBP51 protein was previously found to be associated with influenza virus RNA polymerase [61]. Multiple genes related to RNA splicing and mRNA processing were also differentially expressed in KO (*Ppil1*, *Ppwd1*, *Scnm1*, and *Khsrp* are downregulated, and *Rnps1* is upregulated), suggesting a novel role for *Fkbp5* in transcriptional regulation (Figure 5B). DEGs involved in hydrolysis of carbohydrate may provide insights into the previous association of *Fkbp5* with diabetes [62]. IPA upstream regulator prediction identified several molecules that play roles in the regulation of these DEGs. The Vitamin A derivative Tretinoin was predicted to be activated, while the lipid-lowering drug Bezafibrate, Myc, and PPARα were all predicted to be inactivated upstream regulators. Likewise, many microRNAs were identified as upstream regulators of DEGs (Figure 5E).

To identify those genes with higher fold changes (FC) and greater statistical significance between KO and WT, volcano plots were produced for male (Figure 6A) and female (Figure 6B) comparisons. Some of the most significant genes are labeled on the plot. Of the DEGs with *p*-value < 0.05 and FC > 2, 44 are male-specific and 52 are female specific, with several genes in common. The expression log2 fold changes (FC) of these genes between KO and WT in male and female are listed (Figure 6C). Genes selected for their functional importance, such as microtubule binding activity (*Arl3*), cancer (*Sfi1*) and carbohydrate metabolism (*E4F1*), and their relevance to disease were validated using qRT-PCR, as shown in Appendix A. Most of these genes were confirmed with similar fold changes, with only a few exceptions. The alterations in the gene expression were further validated by the corresponding protein expression levels in certain targets. Peptidylprolyl isomerase domain and WD repeat containing 1 (PPWD1) is a type of PPIase that modulates pathogenic protein multimerization and is related to the processing of capped intron-containing pre-mRNA [63,64]. Folate hydrolase 1 (FOLH1) is a glutamate carboxypeptidase II and an activator of excitatory neurotransmission that regulates long-term memory [65], and is associated with cognitive and social deficits in mice [66]. FOLH1 has multiple substrates, including the neuropeptide N-acetyl-l-aspartyl-l-glutamate (NAAG), and activates the metabotropic glutamate receptor type 3 (mGluR3). Both PPWD1 and FOLH1 proteins were decreased in KO when compared to WT (Figure 6D).

Gene Ontology (GO) analysis was performed to identify DEG functional characteristics, and some top enrichment categories are listed in Appendix A. In agreement with the IPA analysis, genes related to cytoskeleton, plasma membrane, and cell adhesion were identified. Remarkably, the cytoskeleton term has the highest enrichment score, and multiple microtubule function-related genes are differentially expressed between KO and WT (Appendix A). Further annotation clusters highlighted genes in microtubule-associated complexes, microtubule motor activity, and dynein complex; critical in neuronal development and function (Appendix A). Our previous research demonstrated that *Fkbp5* plays an important role in neuronal development and synaptic activity, and we found some initial evidence of its role in microtubule-associated protein regulation [41,43]. The current data support our previous observation that *Fkbp5* is involved in dendrite formation and neuronal signaling, and further identified microtubule-associated genes of interest. Cytoskeleton-associated gene expression was analyzed by qRT-PCR, confirming the downregulation of *Dnaic1*, *Dnah7b*, *Dnah10*, *Kif5a*, *Klc4*, *Tuba1c*, and the upregulation of *Dnah8* (Appendix A).

The top network and its associated molecules are interconnected, which were found to affect neuronal function (Appendix A). This gene network is centered on Akt, Creb, MAPK, and NF𝜅B, with connected genes involved in neuronal function and cell morphology, which were differentially expressed between KO and WT (Figure 6E). Both FKBP5 and Akt are involved in the regulation of depression via dendritic modulation, so Akt and other genes in the network were specifically interrogated. *Fkpb5* KO exhibited upregulation of p-Akt and total Akt, suggesting increased Akt activity (Figure 6F), which is consistent with the findings of others [67]. Some genes in these networks are important for neuronal function, such as NMDA receptor, and protein disulfide isomerase family A member 6 (*Pdia6*); high levels of PDIA6 in Parkinson’s Disease (PD) is associated with greater sensitivity to ER stress [68]. Network member Serpina3 has been directly linked to Alzheimer’s Disease (AD) [69,70]. Annexin A4 (*Anxa4*) functions in glucose-induced cell migration [71] and CALR mediates NFκB activation [72]. Another gene network identified is related to the endocrine system, lipid metabolism, and small molecule biochemistry. This network, centered around P70S6K, HSP90, Vegf, and ERK1/2 highlights the downregulation of PPIase gene clusters (*Ppwd1*, *Ppil1*, *Fkbp4*), and upregulation of *Gask1β*, *Gba2*, *Phgdh* and others (Appendix A). The data support a role for *Fkbp5* in the regulation of metabolism and neuronal activity via the regulation of Akt and MAPK signaling.

### 3.5. Alcohol Affects the Expression of Genes Related to Metabolism and Neuronal Development

Pathway analysis was performed to further understand how acute alcohol stimulation affects biological function and disease of all union DEGs. Union DEGs (422 genes) in both male and female comparisons were used as input and the top Ingenuity canonical pathways are listed in Appendix A. Ketogenesis, dopamine receptor signaling, and serine biosynthesis have the greatest statistical significance. The identification of the insulin receptor signaling pathway indicates that *Fkbp5* plays a role in insulin signaling after acute alcohol, which can alter the activation of PI3K and Akt. When compared to the naïve condition, alcohol treatment predicted some similar IPA functional categories (e.g., lipid metabolism, cell morphology, and post-translational modification), while organismal injury, cell death, and many others were significantly altered by alcohol injection (Appendix A). The data indicate that *Fkbp5* likely plays a role in response to acute alcohol injection and that the dysregulation of brain metabolic functions is a potential mechanism for neuron function-related disease development.

To better understand the common response to alcohol injection, comparisons of 82 DEGs (*p* < 0.05) between alcohol-treated KO and WT mice present in both males and females were used for IPA analysis (Figure 7A). These genes were grouped into four clusters (numbered 1–4) with some of the relevant functions indicated to the right of the heat map (Figure 7B). Genes in clusters 1 and 2 are increased, while genes in clusters 3 and 4 are decreased in KO relative to WT. Some of the genes involved in these functions are listed under the heatmap. These functional clusters indicated differences between KO and WT in response to alcohol injection in the hippocampus. The common DEGs and their involvement in various pathways following alcohol injection in males and females were also discovered (Figure 7C). The top three most significant canonical pathways identified (*p* < 0.001) include methylglyoxal degradation I, methylmalonyl pathway, and serine biosynthesis. Alterations in methylglyoxal degradation I results in changes in the metabolism of amino acids (Arginine and Lysine), lipids, glycated proteins, and glucose. Previous studies suggested that methylglyoxal accumulation induces mitochondrial dysfunction, increases reactive oxygen species (ROS), and impairs neurite extension [73,74,75]. Other enzymes related to mitochondrial function are also differentially affected in KO and WT animals (Figure 7C). Methylmalonyl-CoA mediates the breakdown of certain amino acids and lipids, and is linked to mitophagy [76]. Glutaryl-CoA dehydrogenase (GCDH) is a mitochondrial matrix protein, and its mutation can cause neurometabolic disorder [77]. The 2-oxobutanoate degradation I pathway is also important for mitochondrial function and its downstream metabolite succinyI-CoA is a TCA cycle substrate. Our recent studies support *Fkbp5* playing roles in mitochondrial function and mitophagy (in submission). It is evident that amino acid synthesis (particularly serine and glycine) and degradation (particularly tyrosine and isoleucine) are significantly affected. Energy usage pathways, including sucrose degradation, ketolysis, and ketogenesis were also affected These data provide evidence of a role for *Fkbp5* in the cellular response to alcohol stimulation, primarily through the regulation of mitochondrial function-related pathways and amino acid biosynthesis and degradation.

Alcohol is a chemical insult to the brain and affects its function, particularly within the area of the hippocampus [78,79,80]. IPA analysis identified that the molecular functions most significantly altered by alcohol include the concentration of glutathione, as well as cancer- and inflammation-related functions (Figure 7D). Metabolism related functions, such as synthesis of cholesterol ester, secretion of cholesterol, and assembly of lipoproteins indicate a role for Fkbp5 in the alteration of metabolism within the hippocampus as a response to alcohol injection. Interestingly, many miRNAs are predicted to be upstream regulators (Figure 7E), along with the activators PPARα, pirinixic acid (a PPARα agonist), and mibolerone (a synthetic estrane steroid with high affinity to AR and PR) and the inactivators TNF, dexamethasone, and APOE (lipid metabolism). These results indicate that alteration of these upstream regulators may differentially affect the KO and WT response to alcohol injection. Importantly, manipulation of these upstream regulators could also represent an approach for the molecular manipulation of the DEGs.

Volcano plots depict the DEGs in males (Figure 8A) and females (Figure 8B) with higher fold changes (FC) and greater statistical significance between KO and WT, with some of the most significant genes labeled. Of those genes with significant *p*-value and FC > 2, 35 are male-specific, 53 are female-specific and *Serpina3m* is among the common DEGs (Figure 8C). Common DEGs between males and females were used as input to identify top diseases and functions Appendix A. Again, lipid metabolism and small molecule biochemistry scored highest among the diseases and functions. The top network identified is centered around ERK1/2, P^70^ S6k, and Akt and is circled by the downregulation of PPIase genes (e.g., *PPWD1* and *PPIL1*), which affect protein complex formation and RNA binding. Some neuronal inflammation-related genes (e.g., *TREM2* and *NEO1*) are also downregulated. This network includes the upregulation of genes related to neural function (e.g., *Serpina3* and *PCSK1N*), and lipid metabolism and lipoprotein assembly (e.g., *ACAT2*, *GLO1*, and *MTTP*) (Figure 8D). Thus, the primary differences between the KO and WT hippocampal response to acute alcohol injection include lipid and mineral metabolism pathways, protein and RNA binding, and neuronal function.

### 3.6. Common and Sex-Specific Response to Alcohol Injection

A venn diagram was used to illustrate the number of overlapping DEGs between male and female, between naïve and alcohol, and the combination of these factors. There are 12 genes differentially expressed between all four groups (Figure 9A), with 8 upregulated and 4 downregulated in KO (Figure 9B). Functional analysis showed the enriched GO-Term (Figure 9A). Among the 12 genes, some are known to be important for neuronal function, ion binding, and peptidase activity. For example, Serpina3m is serine (or cysteine) peptidase inhibitor, clade A, member 3M, and has antiapoptotic function by regulating the PI3K/AKT signal pathway. Axna4 is a calcium/phospholipid-binding protein which promotes membrane fusion and is involved in exocytosis. Tagap is a T-cell activation Rho GTPase-activating protein. Dynein heavy chain 8 (Dnah8) is a force generating protein and produces force towards minus ends of microtubules. The log2 FC of the 12 DEGs between all four groups can be found in Appendix A. Some DEGs are specific to each group. For example, there are 14 vs. 13 DEGs in the naïve condition in either female- or male-only groups, respectively, while there are 24 vs. 12 in the alcohol-treated condition in either female- or male-only groups, respectively.

From the heat map of alcohol-treated gene expression (Figure 7B, yellow box), we initially observed that certain genes cluster together in their response to alcohol with a unique trend between sexes. To further investigate the differential gene response to alcohol treatment in male and female, the log2 FC (KO/WT) between the naïve (x-axis) and alcohol-treated (y-axis) conditions were plotted (Figure 9C,D). Each dot represents a DEG with its FC in naïve and alcohol-treated groups. Dots falling into quadrants I or III indicate that the FC difference between the naïve and alcohol-treated conditions is in the same direction. Dots in quadrants II or IV indicate that the FC between KO and WT is in the opposite direction between naïve and alcohol-treated, signifying that the same gene responds differently between KO and WT. Any dot straying away from the 45-degree line is indicative of a higher level of difference between the treatment conditions. The relationships of differences in gene expression between the naïve and alcohol-treated conditions were plotted for males (Figure 9C) and females (Figure 9D) separately, and significant DEGs between treatments were labeled. For some of those genes, such as *Nptxr*, qRT-PCR was performed to confirm the changes in expression. *Nptxr* is involved in LTD in the hippocampus [81] and is expressed in opposing directions in the naïve and alcohol-treated conditions, more so in males (Figure 9E). In other words, the decrease in *Nptxr* gene expression after alcohol is greater in male KO (decreasing 20.9%) than in male WT (3.9%) (Figure 9E) and is less pronounced in female KO (11%) than in female WT (24.22%) (Figure 9F). This result indicates that some genes may respond uniquely to differences in sex as well as genotype. Differences were observed in the male and female responses of several of these genes, including *Gm1821, Rec8*, *Mid1*, *Fcrls*, and *Ccl9*. The expression FC between KO and WT in naïve and alcohol injection conditions can be found in Appendix A.

## 4. Discussion

We previously determined that *Fkbp5* is associated with alcohol drinking and alcohol withdrawal severity [21,22]. The hippocampus plays an important role in learning and memory, and in alcohol consumption. The current study is the first to investigate the differences in hippocampal gene expression between *Fkbp5* KO and WT mice at baseline and following acute alcohol injection. We find that *Fkbp5* is a gene affecting multiple molecular functions and pathways. The most notable diseases and functions affected include (1) cellular assembly and organization, particularly axon guidance signaling in the hippocampus; (2) post-translational modification and protein folding after alcohol injection, particularly those that may affect proteins related to neurological disease development; and (3) cellular injury, lipid metabolism, amino acid biosynthesis and degradation, and fatty acid β-oxidation, processes highly dependent on mitochondrial function. Notably, multiple PPIase genes (*PPWD1*, *PPIL*, and *FKBP4*) were downregulated in KO and were predicted by altered PPIase and isomerase activity. The effect of sex differences in overall gene expression profiling after alcohol treatment was also tested. Several functional studies to understand how *Fkbp5* affects mitochondrial signaling and neuronal morphology after alcohol stimulation were also performed in cultured neurons.

Disease-related pathway and gene network analyses revealed that *Fkbp5* expression is related to neuronal function and disease. Further in-depth analysis indicated that *Fkbp5* regulates microtubule-associated gene expression. Genes related to axon guidance are profoundly affected, suggesting a potential novel function of *Fkbp5*. Axon guidance genes, microtubule function-related genes, and genes related to cation and ion binding, including calcium signaling and calcium transport I, are all essential for neuronal development and function. It is highly probable that FKBP51 protein directly impacts neuron development and function through the regulation of genes in these pathways. Building upon previous findings that FKBP51 protein affects neurite outgrowth, we discovered additional evidence of genes and networks altered in the absence of *Fkbp5*. Axonal transport is responsible for the movement of mitochondria, lipids, and synaptic vesicles to and from synapses, via the motor proteins kinesins and dynein. We find that kinesin-related genes (e.g., *Kif5a*) are decreased in the KO, which could reduce anterograde transportation. Other studies have indicated that KIF5/microtubule interactions regulate γ-Aminobutyric Acid, Type A (GABAA) receptor trafficking in the cortex [82]. Previously, functional GABAergic alterations were observed in the *Fkbp5* KO hippocampus in the form of increased mIPSC frequency and an increase in GABA levels [43]. On the other hand, dynein-related genes (e.g., *Dnah8*) are elevated, which could increase retrograde transportation to the cell body. Several dynein-related genes are up- or downregulated, thus providing a potential explanatory mechanism for the role of *Fkbp5* in neuron development. Recent research revealed that FKBP5 enhances AMPA receptor recycling and affects cognitive function [83]. Given its role in neurite outgrowth and its relationship to pathways involved in neuronal development and function, the exploration of mechanisms by which *Fkbp5* influences such development in normal and diseased conditions should be further investigated [10,84,85,86].

Acute alcohol injection affected multiple amino acid synthesis and degradation enzymes (Methylmalonyl pathway), which participate in fatty acid and amino acid metabolism and generate succinyl-CoA in the TCA cycle. From neuron culture, we determined that *Fkbp5* KO has enhanced mitochondrial signal, which may result in better calcium homeostasis. Indeed, GO analysis identified ion binding, cation binding, metal ion binding, transition and metal ion binding, zinc ion binding as being enriched in both male and female KO with or without alcohol treatment. More than 20% of DEGs are in this functional category, suggesting that the manipulation of *Fkbp5* could result in profound effects on many biological processes beyond mitochondrial function. In addition, alcohol-induced alteration of primary hippocampal neuron morphology is consistent with the identification of cell morphology function alteration in the hippocampus with EtOH treatment. Our approach runs nicely parallel with other research to seek the link between neuronal morphology and global transcriptome alteration; in the future, the application of novel labeling method is a promising approach [87,88]. Alternatively, the identification of gene signature using the KO model or the application of treatment may be other approaches [87,88,89,90,91]. Collectively, these genes affect protein binding, heterocomplex formation, and enzyme activity. The addition of alcohol results in HSP90 and HSP70 activation, further compromising PPIase activity. These multiple avenues of PPIase downregulation represent a significant finding that are differentially regulated in KO after alcohol treatment.

Multiple genes critical for AD were also differentially expressed between KO and WT after alcohol injection. For example, triggering receptor expressed in myeloid cells 2 (*TREM2*) has been directly linked to AD onset [92,93,94]. The altered metabolism of lipids, minerals, amino acids, and others denotes the activation of cellular defenses to alcohol stimulation in the KO. We found that the neuroprotective role of the thimet oligopeptidase (THOP1) pathway was altered after alcohol injection. THOP1, a metalloendopeptidase implicated in the metabolism of bioactive peptides, can be affected by acute cocaine treatment or neurotoxic agent administration [95]. In neurons, THOP1 co-localizes with Aβ and phosphorylated Tau [96], and its expression increases with age [96,97]. FKBP51 and Hsp90 form a mature chaperone complex and prevent Tau degradation [98]. Additionally, the level of FKBP51 protein also increases with age, an event associated with neurotoxic Tau accumulation and AD progression [98]. The significant alterations observed in genes related to THOP1 in KO following alcohol injection is consistent with the finding that these pathways play important neuroprotective roles in AD [96]. Another fascinating discovery is that serine peptidase inhibitor, clade A, member 3M (*Serpina3m*), (also known as alpha-1 antichymotrypsin (ACT)) is the top DEG with high fold change between KO and WT. This gene has been implicated in the pathology of AD, and a study using ACT/APP mice confirmed its role in age-related plaque deposition [69,99,100]. In addition to *Serpina3m*, we found several DEGs that possess serine-type endopeptidase activity or inhibitor function, including *Serpina3f*, also known as protease serine 33 (*Prss33*). It is known that an imbalance in the serpin-protease equilibrium is implicated in many diseases, including Parkinson’s Disease (PD) and AD [101]. The data strongly support that *Fkbp5* affects neuron development and serine-type endopeptidase activity pathways and may be an important target in alcohol drinking-induced and age-related neurodegenerative disease.

This study provides evidence that large number of genes involved in ion, cation, and metal binding are affected by *Fkbp5* KO and alcohol (Appendix A). From DEGs and pathway analysis, calcium-dependent and -related DEGs were identified (Appendix A). For example, *Pkd1*, a regulator of calcium-permeable cation channels, is downregulated in KO. Calcium regulates mitochondrial function and mitochondria regulate calcium dynamics. The mitochondria play important roles in metabolism, energy production, and cell death. Due to the high energetic demands of the central nervous system, recent research suggests that the disruption of mitochondrial activity may underpin various neuronal disease conditions. We speculate that calcium and cation binding activity are profoundly changed in KO and are differentially affected in WT and KO following alcohol insult. KO mice might be better equipped to handle such insults due to enhanced mitochondrial function.

Our gene profiling study of *Fkbp5* KO and WT mouse hippocampus suggests that *Fkbp5* plays critical roles in neuron development, lipid metabolism, and basic cellular functions. These functions are implicated in mental illness and metabolic syndrome. Acute alcohol treatment differentially affects the expression of genes in a genotype- and sex-specific manner, confirming that the *Fkbp5* KO response to alcohol may occur through regulation of lipid and amino acid metabolism, biochemical processes that are important for mitochondrial function.

## Figures and Tables

**Figure 1 cells-13-00089-f001:**
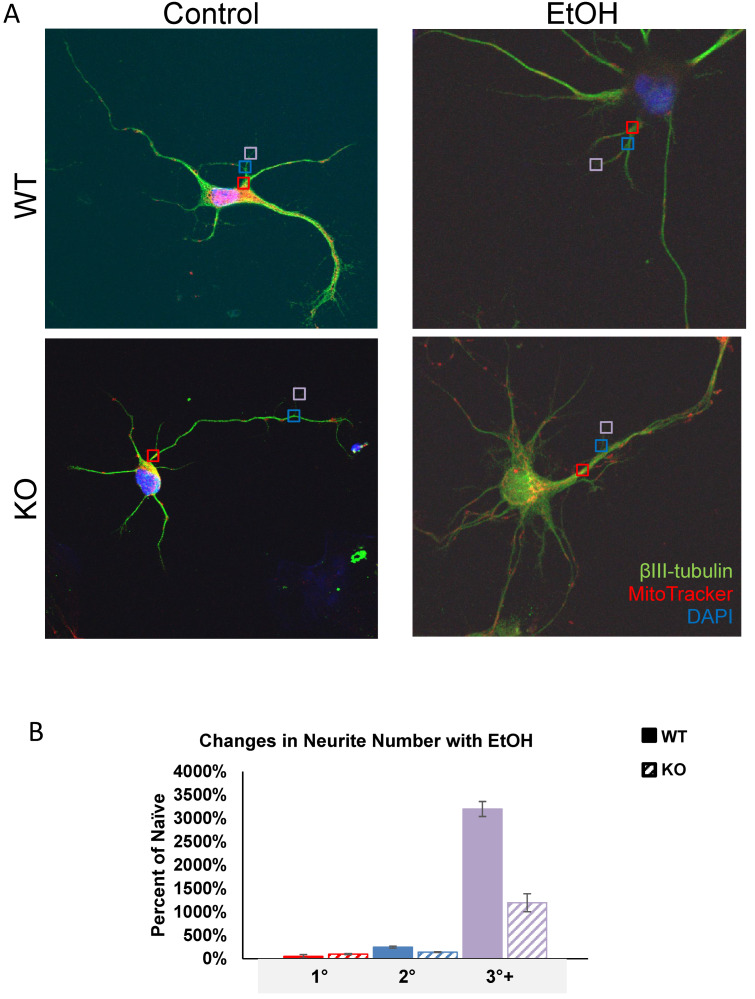
Primary cultured hippocampal neurons with and without EtOH. (**A**) Representative immunofluorescence labeling of βIII-tubulin (for cell body and dendrite), MitoTracker (for mitochondria) and DAPI (for nucleus) expression in WT and KO neurons. Primary (red squares), secondary (blue squares), and tertiary (violet squares) neurites were counted. (**B**) Total neurites from WT and KO naïve and EtOH-treated neurons were counted and the percentages of primary (red), secondary (blue), and tertiary (purple) neurites were calculated for EtOH-treated relative to naïve groups at 3 *DIV*.

**Figure 2 cells-13-00089-f002:**
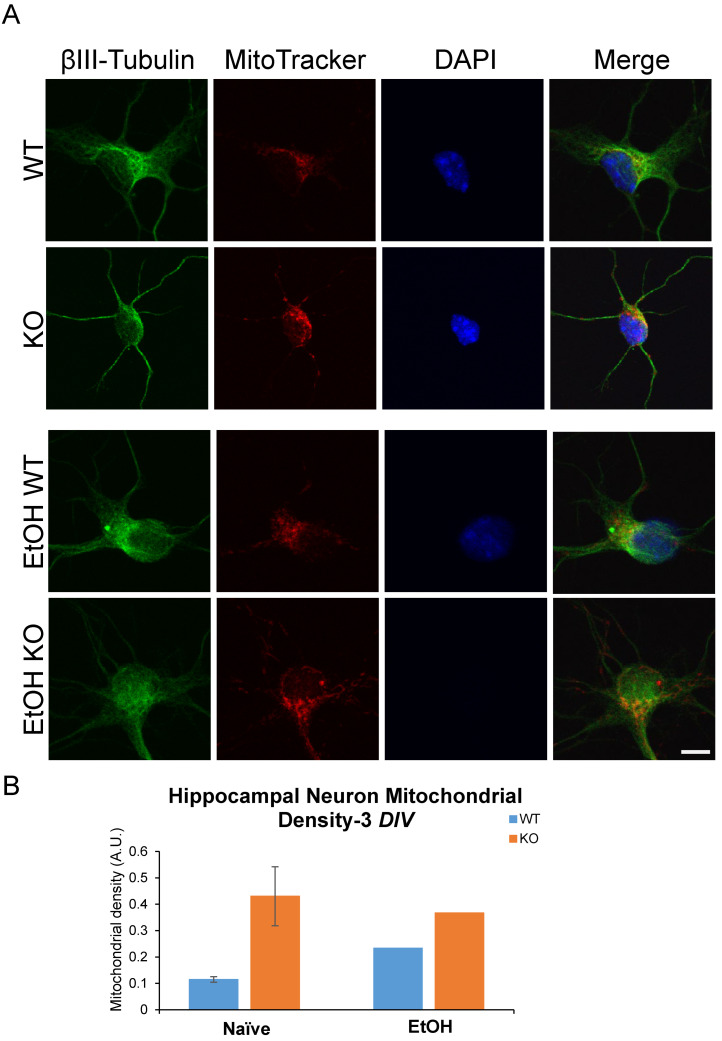
Primary cultured hippocampal neurons with and without EtOH. (**A**) Immunofluorescence labeling of βIII-tubulin, MitoTracker, and DAPI expression in WT and KO neurons at 3 *DIV*. (**B**) MitoTracker intensity in primary cultured hippocampal neurons.

**Figure 3 cells-13-00089-f003:**
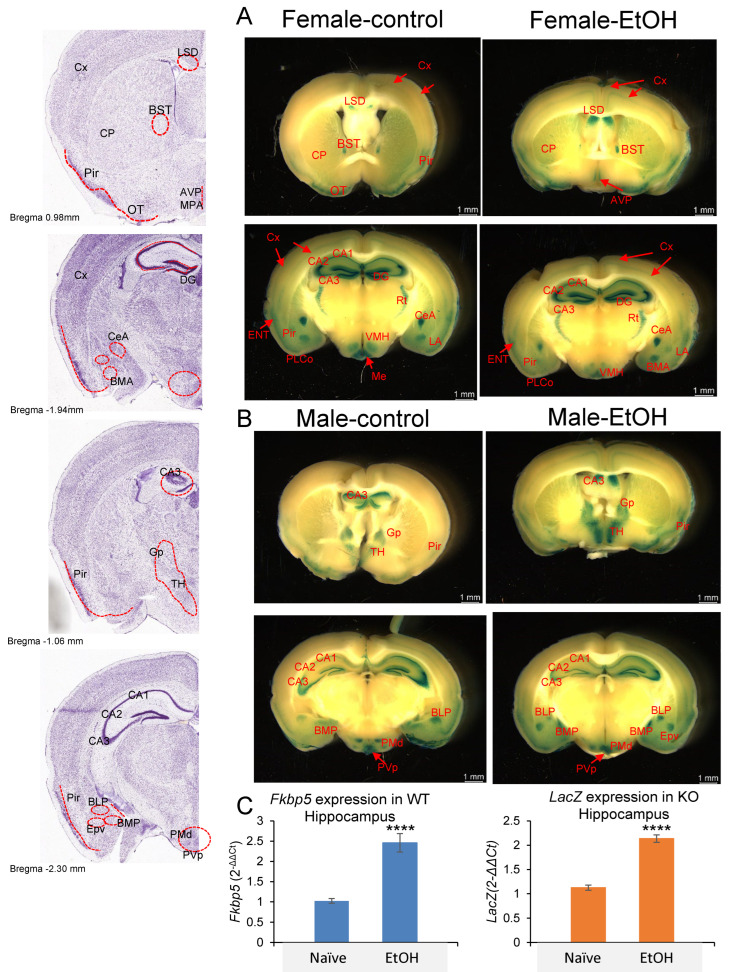
*Lac Z* reporter gene expression in (**A**) female and (**B**) male Naïve and EtOH-injected *Fkbp5* KO mice. Stained anatomical regions noted in red on the left panel. (**C**) *Fkbp5* mRNA expression in naïve and EtOH-injected WT (4 male and 4 female) and *LacZ* mRNA expression in naïve and EtOH-injected KO (4 male and 4 female) hippocampus. **** indicates *p* < 0.0001 by Student’s *t*-test.

**Figure 4 cells-13-00089-f004:**
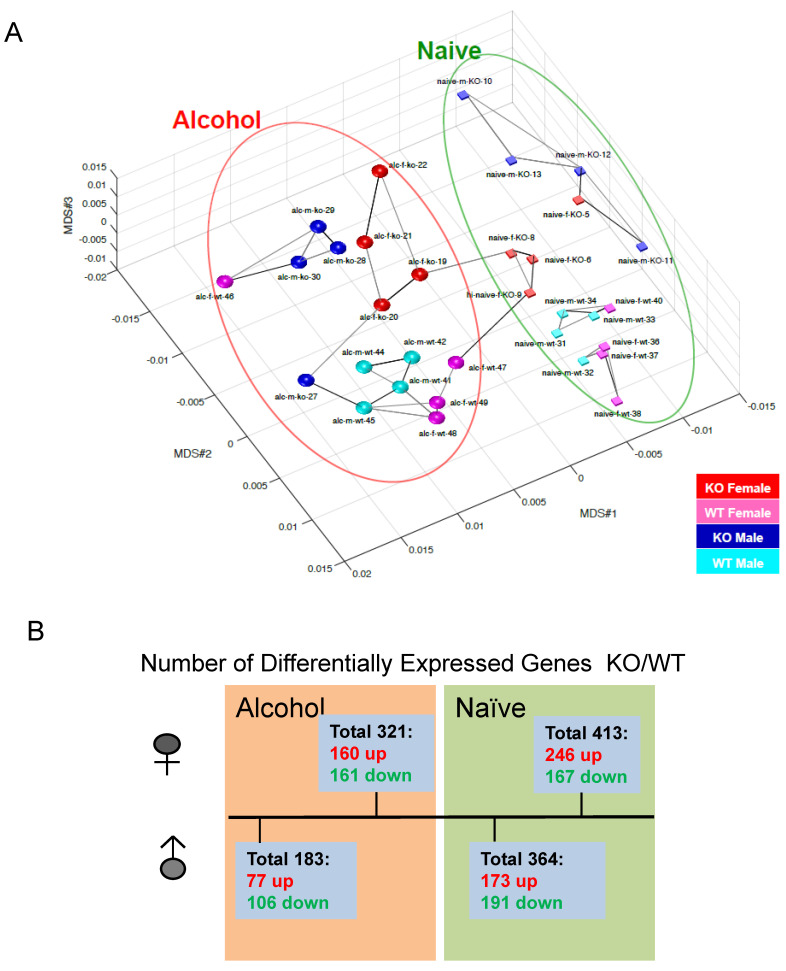
(**A**) PCA detected that naïve and EtOH-treated WT and *Fkbp5* KO expression profiles form distinct clusters. (**B**) Number of DEGs with and without alcohol injection in both sexes, *p* < 0.05.

**Figure 5 cells-13-00089-f005:**
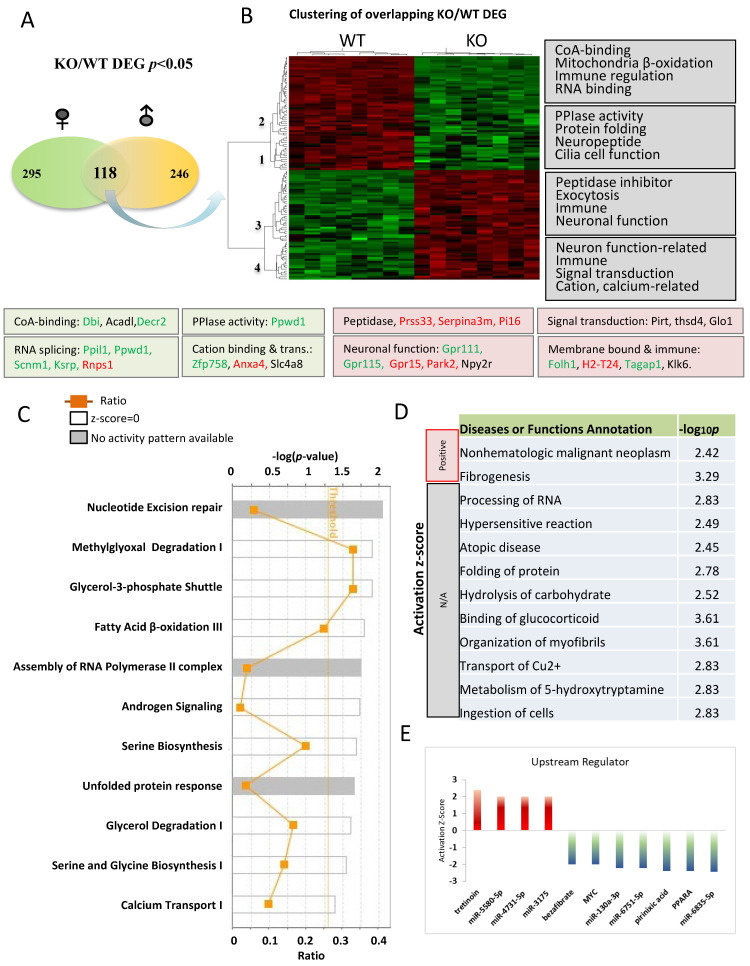
Pathway analysis of DEGs in comparisons between *Fkbp5* KO and WT at naïve condition. (**A**) Number of DEGs identified for male and female. (**B**) Heatmap shows the up- and downregulated genes, which clustered into four groups (functions indicated on the right). (**C**) Canonical pathway analysis of overlapping DEGs genes between male and female. The list of top networks with their scores obtained from IPA including −log (*p-*value) and ratio. The y-axis represents the top canonical pathways as calculated by IPA based on DEGs (union genes) and the x-axis represents the ratio of number of genes from the dataset that map to the pathway to the number of all known genes ascribed to the pathway. The light-yellow line represents the threshold of *p*-value < 0.05 as calculated by Fischer’s test. (**D**) IPA analysis-identified altered diseases and functions. Significance was indicated by −log *p*-value, and activation state were indicated by z-score and others were unknown. (**E**) Upstream regulators prediction.

**Figure 6 cells-13-00089-f006:**
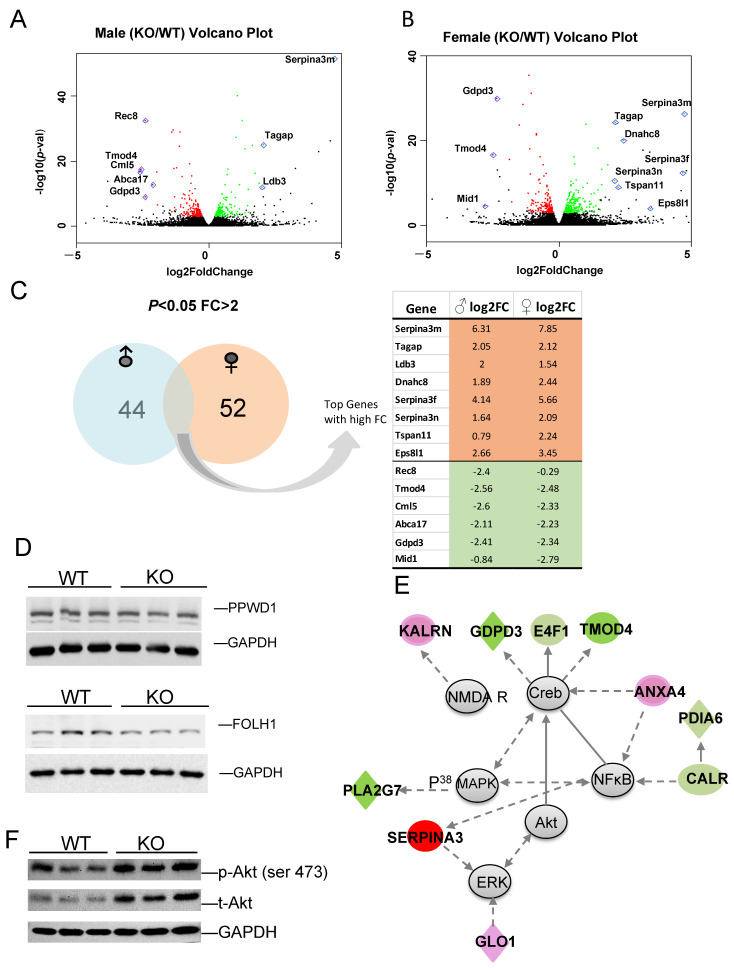
(**A**,**B**) The most significant DEGs displayed in volcano plots for male and female comparisons. (**C**) Number of sex-specific and overlapping genes. (**D**) Protein expression of select targets. (**E**) Representative identified DEG-associated gene network, which functions in endocrine system, development, lipid metabolism, and small molecule biochemistry. Green color represents downregulation, and red color represents upregulation of genes in KO relative to WT. (**F**) KO exhibits higher total and phospho-Akt (Ser 473) relative to WT.

**Figure 7 cells-13-00089-f007:**
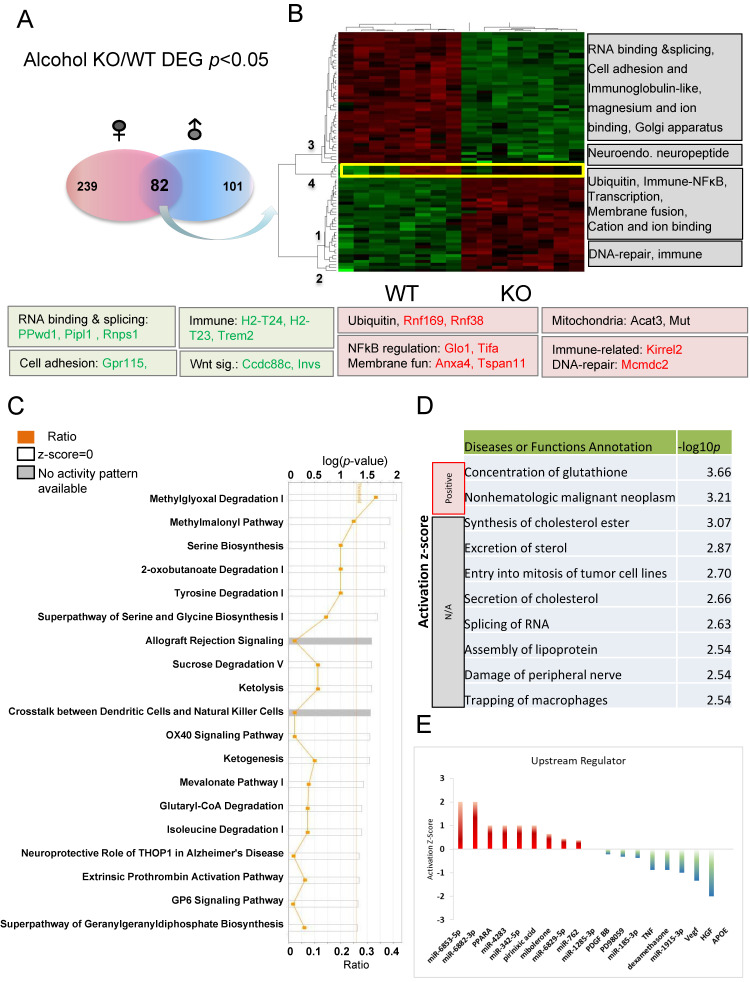
Pathway analysis of DEGs in alcohol-treated comparisons between *Fkbp5* KO and WT. (**A**) Number of DEGs identified for male and female naïve comparisons. (**B**) Heatmap shows the up- and downregulated genes, which clustered into four groups (functions indicated on the right). (**C**) Canonical pathway analysis of overlapping genes following alcohol treatment. The list of top networks with their scores obtained from IPA including −log (*p-*value) and ratio. The y-axis represents the top canonical pathways as calculated by IPA based on DEGs (union genes) and the x-axis represents the ratio of number of genes from the dataset that map to the pathway to the number of all known genes ascribed to the pathway. The light-yellow line represents the threshold of *p*-value < 0.05 as calculated by Fischer’s test. (**D**) IPA analysis-identified diseases or functions. Significance was indicated by −log *p*-value, and activation state indicated by positive or negative mark. (**E**) Upstream regulators prediction.

**Figure 8 cells-13-00089-f008:**
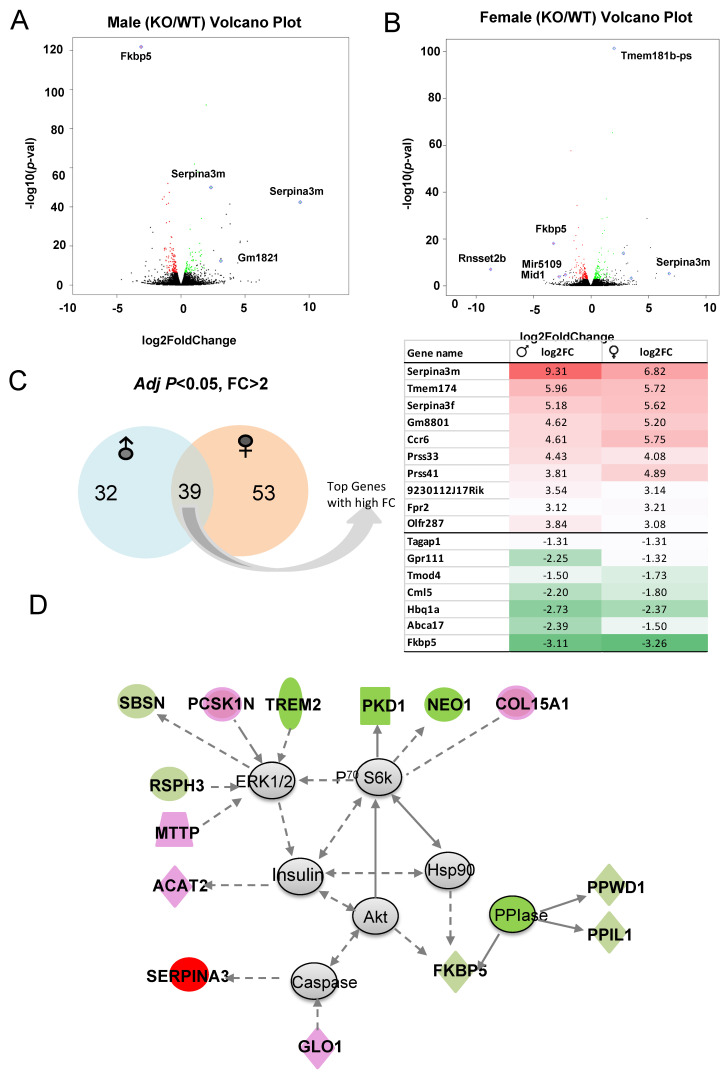
(**A**,**B**) The most significant DEGs displayed in volcano plots for male and female comparisons. (**C**) Number of sex-specific and overlapping genes. (**D**) Representative identified DEG-associated gene network affected by knocking out *Fkbp5* after alcohol treatment. Green represents downregulation, and red represents upregulation of genes in KO relative to WT.

**Figure 9 cells-13-00089-f009:**
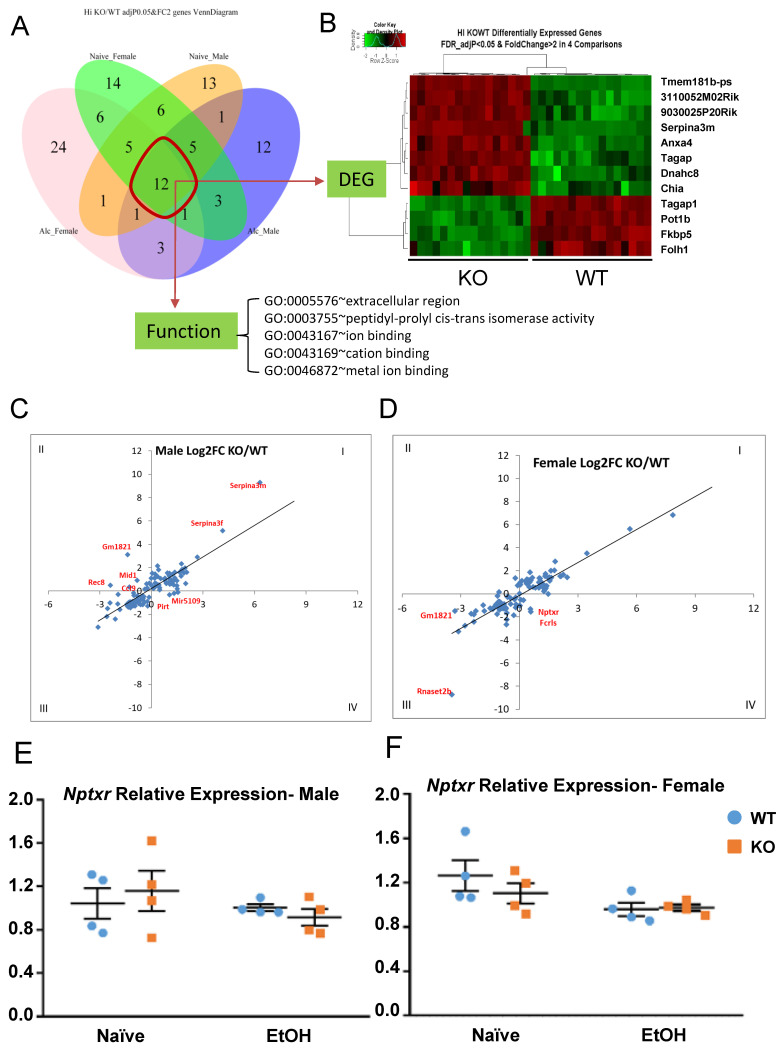
Overlapping genes and sex-specific effects on gene expression in all comparisons. (**A**) Sex-specific, treatment-specific DEGs between KO and WT were plotted. (**B**) Heatmap shows the 12 overlapping DEGs among all groups. (**C**,**D**) Identification of sex-specific responses to alcohol treatment between KO and WT. (**E**,**F**) Relative mRNA expression of *Nptxr* in naïve and alcohol-treated, male and female, KO and WT.

## Data Availability

RNA-seq data have been submitted to a public database (GEO access ID: GSE143354).

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
