# Peer review of "Sex-Specific Impact of Fkbp5 on Hippocampal Response to Acute Alcohol Injection: Involvement in Alterations of Metabolism-Related Pathways"

_cells, 2023, doi:10.3390/cells13010089_

Round 1
Reviewer 1 Report
Comments and Suggestions for Authors
Critical information is missing from the manuscript so it cannot be adequately assessed in current form. Examples include (but may not be limited to the following -
The abstract is missing critical information – for example, results for Mitotracker are reported but it’s unclear what this is a measure of.
Methods:
Section 2.2 should include information about “brain regions” that were dissected, information about the number of mice / litters and the sex of animals included in each primary cell culture experiment.
Section 2.4 - Was BAC measured following alcohol injection, or just inferred from other papers? How old were the animals, what was the sex distribution, and how many litters were represented?
Section 2.5 and 2.6 – how old were the animals and how many litters were represented for each sex?
2.7 – missing specific animal information. Were any controls run?
2.8 – were any controls run?
Information about statistical analysis is missing from this section.
Results – the first paragraph of the results presents information about neurite outgrowth but this is not described in the Methods.
Comments on the Quality of English LanguageLanguage is mostly ok, but should be carefully checked for grammar
Author Response
Reviewer 1
Open Review
(x) Minor editing of English language required
We have revised the paper and reviewed by native English speaker. We have also moved the following sections to the front, which include neuron culture, immunocytochemistry and β-Galactosidase staining sections, to better match the order to result section,
Comments and Suggestions for Authors
Critical information is missing from the manuscript so it cannot be adequately assessed in current form. Examples include (but may not be limited to the following -
The abstract is missing critical information – for example, results for Mitotracker are reported but it’s unclear what this is a measure of.
Answer. In response to the reviewer's comments, we have made multiple revisions to the manuscript and provided additional details and descriptions in the methods and results sections to clarify our findings.
Methods: Section 2.2 should include information about “brain regions” that were dissected, information about the number of mice / litters and the sex of animals included in each primary cell culture experiment.
Answer: We greatly appreciate the reviewer's keen observation. Hippocampus was dissected out and neuron culture was performed. We have repeated the experiment multiple times, however, the sex of the embryonic mice was not able to be distinguished due to their size. Attempts to determine sex by PCR were unsuccessful. We have clarified this fact in the new method section 2.2 Primary postnatal neuron culture.
Section 2.4 - Was BAC measured following alcohol injection, or just inferred from other papers? How old were the animals, what was the sex distribution, and how many litters were represented?
Answer: Thanks for bringing this to our attention. The dosage choice of 2g/kg BW is based on the previously published paper by Dr. Grahame (1). BAC measurement is routinely performed in our lab and we have published the Fkbp5 KO and WT EtOH elimination after acute injection (2), see the figure on the right. Thus, we expected the BAC to reach about 250mg/dL 30 minutes after IP injection, a dosage that produces intoxication and sedation.
Section 2.5 and 2.6 – how old were the animals and how many litters were represented for each sex?
Answer: In section 2.5, adult male Fkbp5 KO (N=16) and female (N=17) mice were used for morphological and histological studies
In section 2.6. Both male and female were used, and each treatment include both male and female, and both genotypes (N=4/sex/genotype/treatment). We have added detailed information to avoid confusion. When mice were sacrificed, they were 3-4 months old. The experimental KO mice were produced by 2 trio breeding schemes, one male was mated with two females. WT were produced from one male and one female mating pairs, and four pairs were used.
2.7 – missing specific animal information. Were any controls run?
Answer: The hippocampus from male WT (N=3) and Fkbp5 KO (N=3) at 8 weeks of age were isolated.
The GAPDH was used as internal control.
2.8 – were any controls run?
The hippocampus from male WT (N=3) and Fkbp5 KO (N=3) at 8 weeks of age were isolated.
The Rpl7 served as the internal control. The outcomes were depicted as the relative fold change, calculated as KO/WT.
Information about statistical analysis is missing from this section.
Answer: This information has been added. GraphPad Prism was used for data analysis (GraphPad Software Inc., San Diego, CA).
Results – the first paragraph of the results presents information about neurite outgrowth but this is not described in the Methods.
Answer: This has been added to the Methods section.
- Grahame NJ, Li TK, Lumeng L. Selective breeding for high and low alcohol preference in mice. Behav Genet. 1999;29(1):47-57.
- Qiu B, Luczak SE, Wall TL, Kirchhoff AM, Xu Y, Eng MY, et al. The FKBP5 Gene Affects Alcohol Drinking in Knockout Mice and Is Implicated in Alcohol Drinking in Humans. Int J Mol Sci. 2016;17(8).
Reviewer 2 Report
Comments and Suggestions for Authors
he manuscript by Kent E. Williams and colleagues reveals a crucial role of Fkbp5 in responding to acute alcohol treatment and its associated pathways through RNA-seq analysis. While the study is both interesting and meaningful, I would like to highlight some concerns about certain limitations.
1, in Fig. 1A, the authors counted the number of 1o, 2o, and 3o neurites, presented with different colors. However, it is somewhat confusing as the meaning of the colors is not explicitly explained. It would be beneficial if the authors could include a description of the color code either within the figures or legends for clarity.
2, in Fig. 2, the interpretation of Mito-tracker intensity is challenging to understand. It would be helpful if the authors clarified whether high-intensity mitochondria correspond to higher mitochondria volume or higher fusion levels. Additionally, it is recommended to include quantification of mitochondria volume in this figure for a more comprehensive understanding.
3, the RNA-seq results are intriguing. To enhance the comprehensibility of the findings, I suggest the authors provide detailed information about the mice used, especially their age. Are the mice in the wild-type (WT) and knockout (KO) groups of the same age? This information would contribute to a better interpretation of the study's results.
Author Response
Reviewer 2
The manuscript by Kent E. Williams and colleagues reveals a crucial role of Fkbp5 in responding to acute alcohol treatment and its associated pathways through RNA-seq analysis. While the study is both interesting and meaningful, I would like to highlight some concerns about certain limitations.
1, in Fig. 1A, the authors counted the number of 1o, 2o, and 3o neurites, presented with different colors. However, it is somewhat confusing as the meaning of the colors is not explicitly explained. It would be beneficial if the authors could include a description of the color code either within the figures or legends for clarity.
Answer: The colors were not chosen for a particular reason other than to differentiate the primary, secondary, and tertiary neurites. We apologize for any confusion.
2, in Fig. 2, the interpretation of Mito-tracker intensity is challenging to understand. It would be helpful if the authors clarified whether high-intensity mitochondria correspond to higher mitochondria volume or higher fusion levels. Additionally, it is recommended to include quantification of mitochondria volume in this figure for a more comprehensive understanding.
Answer: The MitoTracker intensity corresponds to higher mitochondrial density within the neuron. The recommendation to quantify mitochondrial volume is well taken, and future studies can focus on collecting multiple z-stacks to make such quantifications more accurate. Unfortunately for the present study, we do not have the resolution or depth of field to reliably quantify mitochondrial volume.
3, the RNA-seq results are intriguing. To enhance the comprehensibility of the findings, I suggest the authors provide detailed information about the mice used, especially their age. Are the mice in the wild-type (WT) and knockout (KO) groups of the same age? This information would contribute to a better interpretation of the study's results.
Answer: The age of the mice used for RNA-seq should be added in section 2.6. Both male and female were used, and each treatment include both male and female, and both genotypes (N=4/sex/genotype/treatment). We have added detailed information to avoid confusion.
Round 2
Reviewer 1 Report
Comments and Suggestions for Authors
Manuscript has not been revised to address the concerns noted about the lack of detail in the Methods section
Author Response
Answer: The following changes was made to clarify the animal usage and changes were highlighted in the new uploaded version R2.
- In Section 5, Alcohol injection, number of animals used for alcohol injection (8 male and 8female) or naïve (8 male and 8female) were added.
- In Section 6 RNA-sequencing and data analysis, more detailed animal information on breeding of mice was provided, along with number of mice in each treatment, sex, and genotype group. Reads as following.
Mice at the average age 3–4-month-old which were produced from 2 KO breeding trios and 4 WT breeding pairs were used for the study. For no alcohol control (naive) groups, mice were randomly assigned to KO-male -naïve (N=4), KO-female -naïve (N=4), WT-male naïve (N=4), and WT-female naïve (N=4); for alcohol groups, the same number of mice were used, they are KO-male-alcohol (N=4), KO-female -alcohol (N=4), WT-male-alcohol (N=4), and WT-female-alcohol (N=4).
- Figure 3 legend, C. animal number and sex were added for qRT-PCR.
- The whole paper was further revised for English language.
- More discussion was added to discuss neuron morphology and transcriptome discovery.